# Study on the Electrical Conduction Mechanism of Unipolar Resistive Switching Prussian White Thin Films

**DOI:** 10.3390/nano12162881

**Published:** 2022-08-22

**Authors:** Lindiomar B. Avila, Pablo C. Serrano Arambulo, Adriana Dantas, Edy E. Cuevas-Arizaca, Dinesh Kumar, Christian K. Müller

**Affiliations:** 1Departamento de Física, Universidade Federal de Santa Catarina, Florianópolis 88040-900, Santa Catarina, Brazil; 2Department of Chemical and Food Engineering, Federal University of Santa Catarina, Florianopolis 88040-900, Santa Catarina, Brazil; 3Universidad Católica de Santa Maria, Arequipa 04000, Perú; 4Shoolini University of Biotechnology and Management Sciences, Solan 173229, India; 5Faculty of Physical Engineering/Computer Sciences, University of Applied Sciences Zwickau, 08056 Zwickau, Germany

**Keywords:** electrodeposition, Prussian white, resistive switching, electrical impedance spectroscopy, conductive filaments

## Abstract

The electrical conduction mechanism of resistive switching Prussian white (PW) thin films obtained by the electrodeposition method was examined by AC impedance spectroscopy and DC current–voltage measurements. Using an electrode tip to contact PW grown over Au, robust unipolar resistive switching was observed with a current change of up to three orders of magnitude, high repeatability, and reproducibility. Moreover, electrical impedance spectroscopy showed that the resistive switching comes from small conductive filaments formed by potassium ions before the establishment of larger conductive channels. Both voltammetry and EIS measurements suggest that the electrical properties and conductive filament formation are influenced by defects and ions present in the grain boundaries. Thus, PW is a potential material for the next generation of ReRAM devices.

## 1. Introduction

One of the biggest challenges that the electronic devices industry is facing is the requirements of less power consumption, less latency, higher density, higher bandwidth, and lower cost, leading to the development of alternatives such as ReRAMs [1,2]. ReRAMs are considered potential candidates for the next generation of non-volatile memories since they offer high switching speed, simple cell structure, small cell size, high durability, and multistate switching [3,4]. Various types of materials for ReRAM have been proposed, such as transition-metal oxides, perovskite-structured ceramics, and highly insulating oxides including NiO, $TiO_{2}$, ZnO, Cr-doped $SrZO_{3}$, and $SrTiO_{3}$ [5,6,7]. In this sense, finding new materials that are environmentally friendly and have low cost is an important challenge.

Generally, ReRAMs are simple two-terminal devices, forming a metal–insulator–metal (MIM) structure that shows resistive switching, referred as a memristor. Although resistive switching was observed in the mid-1960s on several devices built in the MIM format [8,9], the theoretical modeling of memristors was described in the 1970s [10], and only in 2008, devices were reported [11]. 

Regarding the RS effect, the switching can be classified into two types: bipolar (BRS) and unipolar (URS). In addition to the intricate parameters of each material, other factors that can determine the type are the current and potential limitation window [12,13]. 

Prussian blue analogs (PBAs) are a class of materials with high potential for ReRAMs. The growth of Prussian blue analog (PBA) films has been successfully achieved by electrodeposition [14], liquid-phase epitaxy [15], and spin coating [16]. Among them, electrodeposition is a very effective method for the growth of thin PBA films on a solid substrate due to its high reproducibility, low cost, and ease of controlling deposition parameters (e.g., deposition rate, film thickness, morphology) [14]. Prussian blue (PB) and Prussian white (PW) belong to PBAs. PW is a ferrocyanide complex $K_{x}Fe^{2+}\left[Fe^{2+}{\left(CN\right)}_{y}\right];$ the differences between PB and PW are the oxidation state of Fe ions bound to nitrogen atoms and the insertion of potassium atoms in the structure [17,18,19,20,21].

However, the electrical properties of PBA films are rarely investigated. In previous works, characterizations were presented for Prussian blue (PB) and Prussian white (PW) thin films that showed the behavior of bipolar resistive switching (BRS) and unipolar resistive switching (URS) types, respectively. The samples showed excellent stability of up to 500 cycles and an ON/OFF ratio of three orders of magnitude, with strong evidence that the conductive mechanism occurs in metallic filaments due to the migration of K ions to one of the interfaces [22,23].

The present work shows that electrodeposited PW films can have unipolar resistive switching behavior. By the defined variation of electrical parameters and by studying the PW films with electrical impedance spectroscopy, a deeper analysis of the unipolar resistive switching characteristics and the underlying conduction mechanism is obtained. 

## 2. Experimental Section

### 2.1. Material Preparation

The electrochemical deposition of the PW films was performed in potentiostatic mode using an electrochemical workstation (Ivium CompactStat, Eindhoven, Netherlands), at room temperature by using Au/Cr/Si substrates as working electrodes, a *Pt* foil as a counter electrode, and a saturated calomel electrode (SCE) as reference (Figure 1a). All the voltages in the text refer to this reference electrode. The working electrode was prepared by evaporating 50 nm Au on 5 nm Cr on (100) Si substrates with a size of 1 cm×1 cm at a base pressure of $10^{-5}$ Pa (Figure 1b). The deposition of PW occurred in a circular area of $0.5\;cm^{2}$ defined by a mask of adhesive tape on the surface of the working electrode. As electrolyte for the electrochemical synthesis, a solution of $0.25\;mM\;K_{3}Fe{\left(CN\right)}_{6}$, $0.25\;mM\;FeCl_{3}$, 1.0 M KCl, and 5.0 mM HCl  at pH 2 was used [24]. All chemicals used were of high purity, >99%, (Sigma Aldrich, Darmstadt, Germany). The PW layers were deposited by applying a constant potential of 0.1 V at 25 °C and limiting the electrodeposited charge to 30 mC. 

### 2.2. Characterization

The morphology of the PW films was investigated with a TESCAN field emission scanning electron microscope (FEG-SEM, TESCAN CLARA, Brünn, Czech Republic) at 10 kV.

Electrical characterization (current vs. voltage (I×V)), was performed on electrodeposited samples using an electrochemical workstation (Ivium CompactStat, Eindhoven, Netherlands) with a dc voltage ranging from −1 to+1 V. For the electrical measurements, a 2-point probe was used with one contact at the top of the PW film using a metallic tip with surface area of $0.5\;mm^{2}$ and the Au on bottom layer (Figure 1b). Electrochemical impedance spectroscopy (EIS) was performed in an impedance analyzer (Agilent 4294A, Santa Clara, CA, USA) operating in the frequency range between 40 Hz and 100 MHz. The applied potentials were AC $V_{RMS}\;$ of 100 mV and fixed DC bias voltages of 0.5 V, 1.5 V, and 2.0 V.

## 3. Results and Discussion

### 3.1. Morphological Characterization

Figure 2 shows a cross-section image of the PW film. From the electron microscopy image, an average film thickness of about 500 nm can be determined. Surface analysis of the PW film with atomic force microscopy (not shown) gave an average peak-to-valley height of approximately 100 nm. The morphology with different grains is visible along the whole layer and shows cubes along the layer thickness. The preferential growth direction of the majority of the cubes is [111]. Generally, the grains near the bottom are significantly smaller in size than the grains at the top. Such a grain distribution and the appearance of cubes typically occurs by the Volmer–Weber growth process, where three-dimensional islands are formed on the substrate surface and then grow in sequence until they coalesce into a continuous film. Our films prove that a good control of film thickness, morphology, and structure can be achieved by electrodeposition. This result makes electrodeposition attractive against other film deposition methods, such as spin coating or liquid-phase epitaxy.

### 3.2. Electrical Characterization

Figure 3a shows the result of the electrical measurement in a sandwich structure with the Au layer as bottom contact and a metallic tip as top contact. The voltammetry was performed between −1  and  1 V. The current was limited to 1 mA to avoid damage [25,26,27]. In the bias voltage between 0 and 1.0 V, the high-resistance state (HRS), off state, switches at 0.7 V (step 1) to the low-resistance state (LRS), the on state. The LRS is maintained up to 1.0 V and is reset into HRS at 0.3 V when the current abruptly decreases (step 2), back to the off state. Similar behavior is observed in the negative-bias region, where the switch occurs at around 0.9 V (step 3) and the resetting at 0.4 V (step 4). The resistive switching at the same potential that occurs in the PW films is characteristic for unipolar resistive switching (URS) [28]. Figure 3b presents the same I×V curve on a semi-logarithmic scale showing a current switch greater than three orders of magnitude for both polarities. The curve clearly exhibits a characteristic butterfly shape or the pinched hysteresis characteristics of resistive switching devices [28,29].

The repeatability of the working resistive switching over one hundred cycles is shown in Figure 4a. The on/off ratio between the LRS and HRS states remained around three orders of magnitude in current even after 100 cycles, and the URS effect was preserved for both polarities. This result confirms the reversibility and reproducibility of the PW films required for applications in ReRAM memory devices. In comparison with other materials showing the resistive switching effect reported in the literature (in most cases oxides), we found typical switching ratios in the range from 10 to 10^6^ and endurances in the range from 1 up to 10^7^ [30]. However, it is important to note that most of the studied materials show values comparable with our PW samples. The conduction mechanism that occurs in the PW thin film was analyzed in the HRS (off state) and positive-bias region, as shown in Figure 4b. The  I×V  curves of the 1st, 25th, 50th, and 100th scans were plotted with a log–log scale and the data were fitted. For the lower applied field at HRS region, up to around 0.3V, the slope values varied between 0.5 and 0.74 indicating a high resistive state. Further increment in the applied electric field caused a gradual increment in the slopes close to one with increasing number of cycles, indicating a linear ohmic behavior (I ∝ V) and the formation of small conductive metallic filaments between the top and down electrodes, still in the HRS region [31,32,33,34,35]. Higher slopes with values closer to two can be an indication for a space charge limited current (SCLC) mechanism following a quadratic relationship between voltage and current [36]. However, in our samples, SCLC was not the dominating effect in the investigated voltage range.

To elucidate the electrical response of our samples to the electric field, different voltage scan rates were applied, without increasing the potential range to avoid damage to the PW film. Figure 5a shows the electrical response in the positive-bias range for the system with voltage scan rates of 100, 50, 15, and 0.5 mV/s. Notice that the resistivity switching effect is preserved, but a shift of the setting potential ($V_{set}$) occurs. As expected, when the scan speed decreases, the $V_{set}$ also decreases due to the increased time in which the PW is stimulated by the external potential. The inset in Figure 3a shows the relation of $V_{set}$ and voltage scan rate. 

Figure 5b presents the HRS state for different scan rates on the log–log scale. The angular coefficients for the different voltage scan rates were around 1.17 ± 0.07, indicating an ohmic behavior for the high-resistance state at lower scan rates. At higher potentials, larger filaments are formed, which are responsible for the strong change in the electrical current observed in the film. The filament model suggests that the filaments are composed and formed from defects such as vacancies or metal ions [31]. PW films are rich in potassium ions, and our data strongly support a mechanism based on the generation of filaments induced by the migration of these cations during the formation process, as indicated in Figure 5c. When applying a bias voltage, the positively charged potassium ions can move to the negative pole (top or bottom depending on the polarity). At applied voltages larger than $V_{set},$ the formation of a conductive path can occur, and the resistance drops. According to the filament model [32], the defect lines form tiny conductive filaments in the HRS, and these filaments come together to form larger and more conductive filaments, leading to the transition to the LRS. Defects, grain boundaries, and lattice distortion provide easy diffusion paths for ions, thus forming conduction paths. The results presented here agree with earlier findings obtained from Prussian blue (PB) and Prussian white (PW) thin films [22,23], which indicated that the filamentary conduction mechanism can explain the huge resistance change in those films. 

Thus far, the conductive channels formed by the migration of potassium have not been proven, but the results presented in this work show strong evidence of how and where changes in the material arise upon the application of an external electric field. This hypothesis is due to the fact that thin PW films are potassium rich and these ions are the only free ions in the crystalline lattice and grain boundaries to cause the RS effect. 

For further clarification of the underlying electrical transport mechanism in the PW film, electrical impedance spectroscopy was used. Figure 6 shows the Nyquist plot as well as a schematic diagram of the intermediary filament formation and the respective equivalent circuits used to fit the curves. 

The three Nyquist diagrams show single semicircles fitted by two constant-phase elements ($CPE_{1}$ and $CPE_{2}$) in parallel with the resistances $R_{1}$ and $R_{2}$. CPEs are assigned to the interfaces filament/PW film and PW film/Au electrode that behave as double layers, as shown in Figure 6b. At high frequencies, R2//CPE2 is assigned to interface the Au/PW film and the charge transport in the PW crystals, while at lower frequencies, R1//CPE1 is assigned to charge transport on the interface filament/grain boundary defects with traps and charge accumulation [37,38,39,40,41,42,43]. 

The curves show that lower impedances at high frequencies (left end of the curves) may be explained by Koop’s theory, which assumes that the conductive grains are separated by resistive layers of grain boundaries [44]. In our case, the PW crystals are the grains involved by grain boundaries. The high impedances at lower frequencies (right end of the curves) are related to grain boundaries. 

The series resistance ($R_{0}$) present in all circuits shows larger values for higher DC biases: R_0_ = 173.8, 270.2, and 331.3 Ω for bias 0.5, 1.5, and 2.0 V, respectively. These results strongly indicate the formation/increment of small metallic conductive filaments between both electrodes [45]. On the other hand, as the impedance module is reduced for larger DC bias, and the parallels R/CPEs under 1.5 and  2.0 V present lower impedance values, compared to ones from 0.5 V bias.

The impedance results suggest that the electrical responses are mainly influenced by the formation of conductive filaments, most possibly at the grain boundaries and different interfaces (film/electrodes), in agreement with literature results [22,37,38,39,40,41,42,43] and electrical characterization at DC measurements. The interfaces and respective R//CPE circuits are hypotheses as the physico-chemical origin of such CPE behavior at these interfaces is not yet fully understood [46]. Heterogeneities on the interface have been suggested to lead to a distribution of charge accumulation time scales, due to different times of charge carrier relaxation in these materials, as given by the Maxwell–Wagner two-layer model, in agreement with Koop’s phenomenological theory [37].

## 4. Conclusions

We observed the resistive switching mechanism in electrodeposited layers of Prussian white. The PW films showed a unipolar switching effect with an on/off current ratio of up to three orders of magnitude and good reproducibility of 100 cycles. The electrical measurements show the rise of the ohmic conduction model in the HRS region that suggests the formation of small conductive filaments before the establishment of larger conductive channels for setting the film into the LRS. Both voltammetry and EIS measurements suggest that the electrical properties and conductive filaments formed in the films are determined by the influence of defects and ions present on the grain boundaries. This recent discovery strongly indicates that the conduction channels are formed by potassium ions, the most credible for the resistive switching effect in PW films. This result opens further possibilities such as ionic control in the interstices of PW by using an external electric field and ensuring that the material does not degrade with time and use. However, to access practical applications of PW as next-generation ReRAM devices, the electrical switching characteristics, influence of fabrication parameters, long-time behavior, and impact of environmental conditions need to be carefully studied in future works.

## Figures and Tables

**Figure 1 nanomaterials-12-02881-f001:**
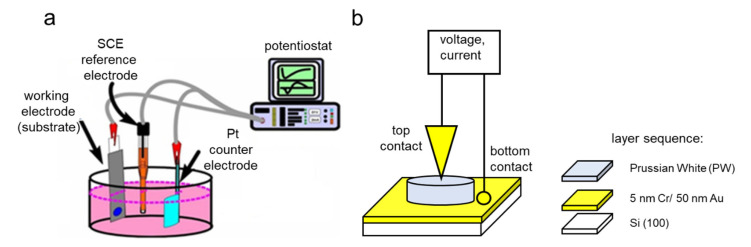
(**a**) Experimental setup for the electrochemical deposition of PW. (**b**) Layer sequence of the samples and applied electrode configuration for electrical measurements.

**Figure 2 nanomaterials-12-02881-f002:**
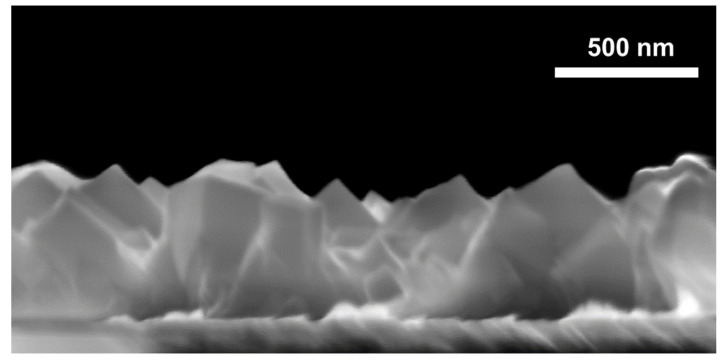
Cross-section electron microscopy image of the PW film.

**Figure 3 nanomaterials-12-02881-f003:**
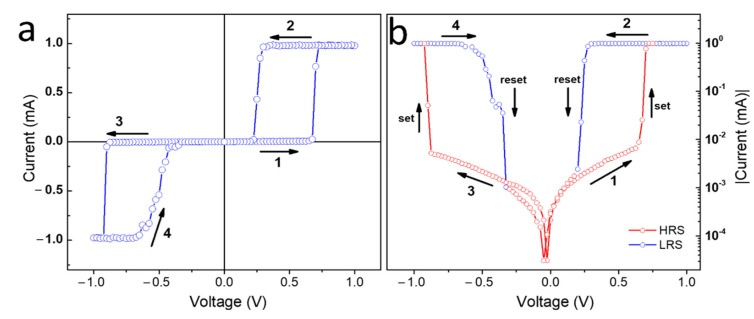
(**a**) *I × V* hysteresis showing unipolar resistive switching characteristics for the PW thin film. (**b**) Logarithmic plot of absolute current over voltage. The red curve represents HRS and the blue curve LRS.

**Figure 4 nanomaterials-12-02881-f004:**
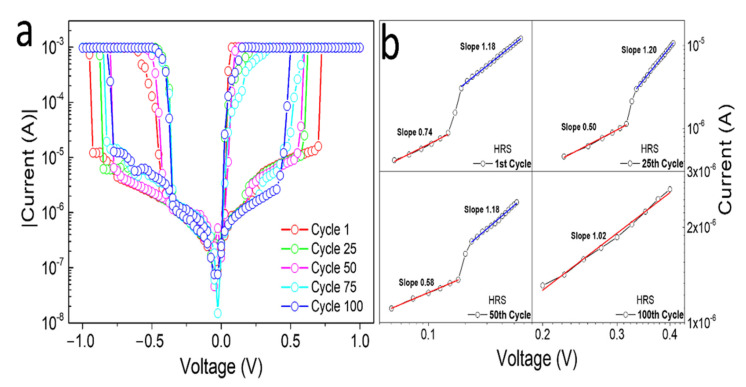
(**a**) *I × V* curves of PW samples during 100 cycles. (**b**) Log—log scale of *I × V* curves in the HRS region in the positive bias voltage for 1st, 25th, 50th, and 100th cycles.

**Figure 5 nanomaterials-12-02881-f005:**
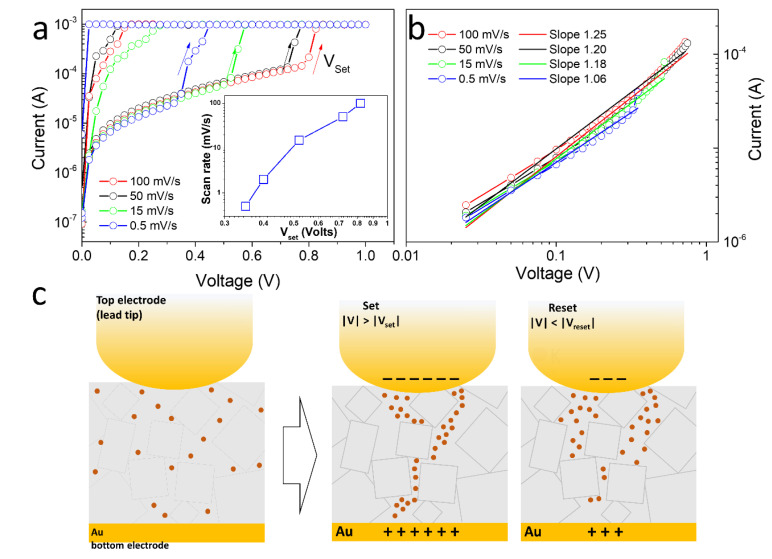
(**a**) *I × V* curves in the positive bias voltage range measured with different potential scan rates. The inset shows the $V_{set}$ as a function of the potential scan rate. (**b**) Log—log scale curves of the HRS state, below $V_{set}$ (**c**) Schematic diagram of metallic filament formation of defects among crystals and free ions dispersed in the PW film.

**Figure 6 nanomaterials-12-02881-f006:**
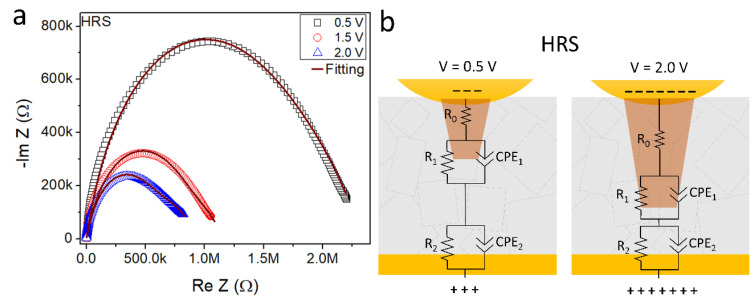
(**a**) Nyquist diagrams of PW/Au sample in HRS (off state) at 0.5, 1.5, and  2.0 V DC bias. (**b**) The corresponding equivalent circuit and schematic diagram of the main interfaces and transport mechanisms.

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
