# Peer review of "Study on the Electrical Conduction Mechanism of Unipolar Resistive Switching Prussian White Thin Films"

_nanomaterials, 2022, doi:10.3390/nano12162881_

Round 1

Reviewer 1 Report

There are however a few issues that the authors are kindly requested to take into account for a revised version:

-       It will be much better if the author can provide more details about the advantages or disadvantages of your method compared with the others.

-       Some discussion would be welcome about the formation mechanism of the films.

-       The substrate for deposition was (100)Si . Provide some structural and morphological characterization of the films? For example: What is the peak-to-valley from AFM imagine?

-       Authors have validated the physical properties of thin film deposited using the technique. They should report and compare the film formed using their technique is either good or bad with respect other films prepared using other direct or indirect techniques.

 Finally, the language is in general correct, but some parts require a revision.

Author Response

Dear Referee,

we thank you for your valuable comments and suggestions and revised our manuscript.

Referee

  1. It will be much better if the author can provide more details about the advantages or disadvantages of your method compared with the others.

We have included a new paragraph (line 48-53), comparing our method with another method suitable for the fabrication of PW. This point is later shortly discussed in the discussion part (line 111-114)

  1. Some discussion would be welcome about the formation mechanism of the films.

To address this point we included a new section dealing with the film morphology. (line 100-116)

  1. The substrate for deposition was (100)Si. Provide some structural and morphological characterization of the films? For example: What is the peak-to-valley from AFM imagine?

We added data from electron microscopy showing the morphology of the film, film thickness, film thickness variation and detail about the growth process. (see line 100-116)

  1. Authors have validated the physical properties of thin film deposited using the technique. They should report and compare the film formed using their technique is either good or bad with respect other films prepared using other direct or indirect techniques.

We discussed the fabrication process in relationship with other methods (line 111-114) and discussed the electrical results in comparison with the literature (line 139-143).

  1. Finally, the language is in general correct, but some parts require a revision.

We checked the language, corrected the figure captions, and added some important references

Reviewer 2 Report

Comments:

The authors studied the electrical conduction mechanism of unipolar resistive switching Prussian white thin films using impedance spectroscopy. Even though the author performs various analysis technique to characterize the unipolar switching mechanisms, it is some missing about the information of device structure and underlying mechanism. Thus, the author is required to provide further explanations with additional information.

#1: In materials preparation section, the author explains the procedure to make the resistive device. However, it is not sufficient to understand the device structure and film thickness. The author is required to provide the information about the each film thickness and electrochemical deposition scheme image of PW film process.

#2: In Fig. 2b, the log-log scale of I-V curves was shown. And the conduction mechanism varies from resistive to ohmic depending on the slope. Is there any supporting equation or relation to estimate the slope and conduction mechanism?

#3: In Fig. 2, What is the maximum cycle number of your device. It seems to be short as cycle 100 to apply this device to real device applications.

#4: In Fig. 3(c), some more detailed explanation is required. The potassium ion is moving from bottom Au electrode to top contact electrode? It is confusing when the bias is applied. The potassium ion is positive ion, so I think you should mention the ion direction and what is K ion.

Author Response

Dear Referee,

we thank you for your valuable comments and suggestions and revised our manuscript.

Referee

  1. In materials preparation section, the author explains the procedure to make the resistive device. However, it is not sufficient to understand the device structure and film thickness. The author is required to provide the information about the each film thickness and electrochemical deposition scheme image of PW film process.

To illustrate that we have included a new figure showing the experimental setup and the device/measurement configuration (see Figure 1).

Film thickness analysis was performed by scanning electron microscopy (see Figure 2) and discussed (line 100-117).

  1. In Fig. 2b, the log-log scale of I-V curves was shown. And the conduction mechanism varies from resistive to ohmic depending on the slope. Is there any supporting equation or relation to estimate the slope and conduction mechanism?

Slopes close to 1 are associated with the ohmic law (I V).  Slopes close to 2 are related to the space charge limited current (SCLC) mechanism (I V²). We extended the discussion about the conduction mechanism (line 147-154).

  1. In Fig. 2, What is the maximum cycle number of your device. It seems to be short as cycle 100 to apply this device to real device applications.

Up to now for PW films we performed up to several hundred cycles. Most of the works in the literature prove the endurance applying 50-1000 cyles. We agree that towards practical applications the long-time behavior and other properties needs to be analyzed (see line 241-244).

  1. In Fig. 3(c), some more detailed explanation is required. The potassium ion is moving from bottom Au electrode to top contact electrode? It is confusing when the bias is applied. The potassium ion is positive ion, so I think you should mention the ion direction and what is K ion.

We added this detail to our discussion and mentioned the direction of motion of the K+ions (line 174-175)

In addition we checked the language, corrected the figure captions and added some important references

Round 2

Reviewer 2 Report

I strongly accept this form for publication